# Correlation between Hospital Volume of Severely Injured Patients and In-Hospital Mortality of Severely Injured Pediatric Patients in Japan: A Nationwide 5-Year Retrospective Study

**DOI:** 10.3390/jcm10071422

**Published:** 2021-04-01

**Authors:** Chiaki Toida, Takashi Muguruma, Masayasu Gakumazawa, Mafumi Shinohara, Takeru Abe, Ichiro Takeuchi, Naoto Morimura

**Affiliations:** 1Department of Disaster Medical Management, The University of Tokyo, 7-3-1 Hongo, Bunkyo-ku, Tokyo 113-8655, Japan; molimula-tky@umin.ac.jp; 2Department of Emergency Medicine, Graduate School of Medicine, Yokohama City University, 4-57 Urafunecho, Minami-ku, Yokohama 232-0024, Japan; mgrmtks@gmail.com (T.M.); gakumazawa-tuk@umin.ac.jp (M.G.); shinoharamafumi@yahoo.co.jp (M.S.); abet@yokohama-cu.ac.jp (T.A.); takeqq@yokohama-cu.ac.jp (I.T.)

**Keywords:** pediatric patient, severely injured patient, volume–outcome relationship, hospital volume, centralization

## Abstract

Appropriate trauma care systems, suitable for children are needed; thus, this retrospective nationwide study evaluated the correlation between the annual total hospital volume of severely injured patients and in-hospital mortality of severely injured pediatric patients (SIPP) and compared clinical parameters and outcomes per hospital between low- and high-volume hospitals. During the five-year study period, we enrolled 53,088 severely injured patients (Injury Severity Score, ≥16); 2889 (5.4%) were pediatric patients aged <18 years. Significant Spearman correlation analysis was observed between numbers of total patients and SIPP per hospital (*p* < 0.001), and the number of SIPP per hospital who underwent interhospital transportation and/or urgent treatment was correlated with the total number of severely injured patients per hospital. Actual in-hospital mortality, per hospital, of SIPP patients was significantly correlated with the total number patients per hospital (*p* < 0.001,). The total number of SIPP, requiring urgent treatment, was higher in the high-volume than in the low-volume hospital group. No significant differences in actual in-hospital morality (*p* = 0.246, 2.13 (0–8.33) vs. 0 (0–100)) and standardized mortality ratio (SMR) values (*p* = 0.244, 0.31 (0–0.79) vs. 0 (0–4.87)) were observed between the two groups; however, the 13 high-volume hospitals had an SMR of <1.0. Centralizing severely injured patients, regardless of age, to a higher volume hospital might contribute to survival benefits of SIPP.

## 1. Introduction

Injury is a major cause of child mortality worldwide and developing appropriate trauma care systems, suitable for children, is desirable [1,2]. Children have age-dependent anatomical and physiological differences compared to adults; therefore, it remains unclear whether or not adult-oriented trauma systems are effective in predicting mortality in severely injured pediatric patients (SIPP) [2,3,4,5,6].

Previous studies have reported that high hospital volumes of severely injured patients were associated with low mortality [7,8,9]. Centralization of a hospital specialized in trauma care was promoted in some developed countries; however, the requirements for a clearly defined hospital volume of severely injured patients for obtaining a maximum mortality benefit remain unclear [8,9,10].

In Japan, with a view toward improving trauma care quality, the Japanese Association for the Surgery of Trauma recommended, in 2009, that trauma centers require a minimum of 1200 injured patient admissions and 150 severely injured patient admissions with an Injury Severity Score (ISS) of ≥16 [11]. However, trauma care systems specialized in treating pediatric patients, such as pediatric trauma centers, have not yet been established, and many injured patients are transported to the nearest tertiary emergency medical centers. One previous study that used a Japanese nationwide trauma database reported that a high hospital volume of severely injured patients with ISS ≥16 reduced the in-hospital mortality; however, there were some limitations to the definition of a high-volume hospital, wherein the definition was an admission of >100 severely injured patients [9]. 

Thus far, there has been no report that evaluated the correlation between the hospital volume of severely injured patients (with ISS ≥16) and in-hospital mortality of SIPP, which has a benefit for in-hospital mortality of SIPP. This study thus aimed to evaluate the correlation between the annual total hospital volume of severely injured patients and in-hospital mortality of SIPP.

## 2. Materials and Methods 

### 2.1. Study Setting and Population

This retrospective observational nationwide study was conducted based on data obtained from the Japan Trauma Data Bank (JTDB), which registers data of patients with an injury and/or burn, and records prehospitalization- and hospital-related information. The JTDB includes data on demographic characteristics, comorbidities, injury types, mechanism of injury, means of transportation, vital signs, Abbreviated Injury Scale (AIS) score, prehospital/in-hospital procedures, injury diagnosis as indicated by the AIS, and clinical outcomes. In most cases, physicians trained in AIS coding record the online registration of individual patients’ data. There was a total of 280 participating hospitals in all 47 prefectures in Japan, including 92% of the Japanese government-approved tertiary emergency medical centers in March 2019. The Japan Association for the Surgery of Trauma permits open access and updating of existing medical information, and the Japan Correlation for Acute Medicine evaluates the submitted data [12].

In this study, we used the JTDB dataset that included information from 1 January 2014 to 31 December 2018, which initially yielded the data of 181,971 patients. The inclusion criteria for this study were severely injury patients with an ISS of ≥16. Patients with burns, cardiac arrest upon hospital arrival, or missing key data were excluded from this study. Figure 1 presents a flow diagram of the patients’ disposition in this study. 

### 2.2. Data Collection

We collected information including the following variables from the JTDB: demographic characteristics (age (years), sex, year of hospital admission, injury mechanism, transportation type, transfer process), clinical parameters (AIS of the injured region, ISS, Revised Trauma Score (RTS), Trauma and Injury Severity Score (TRISS), requirement of urgent treatment; urgent blood transfusion within 24 h from hospital arrival, urgent transcatheter arterial embolization (TAE), and/or urgent surgery), and outcomes (in-hospital mortality, standardized mortality ratio (SMR)). Outcome measures were in-hospital mortality and SMR per hospital. The predicted mortality was calculated using the TRISS, and SMR was calculated by dividing the actual in-hospital mortality by mean predicted mortality.

### 2.3. Ethics Statement

This study was approved the hospital ethics committees of Yokohama City University Medical Center (approval no. B170900003). The approving authority for data access was the Japanese Association for the Surgery of Trauma (Trauma Registry Committee). The requirement of informed consent from the patients was waived owing to the observational nature of the study.

### 2.4. Statistical Analysis

The primary analysis estimated (1) the correlation between the number of total and pediatric patients with severe injury, per hospital; (2) the correlation between the total number of severely injured patients of SIPP who underwent interhospital transportation and/or urgent treatment per hospital; and (3) the correlation between the total number of severely injured patients and in-hospital mortality of SIPP, per hospital. The secondary analysis compared the clinical parameters and outcomes, per hospital, between low-volume hospitals and high-volume hospitals. Hospitals were divided into two groups according to the distribution of severely injured patients and volumes were calculated as average annual values over a 5-year study period: high-volume, ≥150 severely injured patients/year; low-volume, <150 severely injured patients/year admission, at each hospital. Admission of >150 severely injured patients with an ISS of ≥16 per year was one of the requirements which the Japanese Association for the Surgery of Trauma recommended for a trauma center. 

Continuous variables are presented as median with ranges (minimum–maximum), and categorical variables are presented as the number and percentages of patients. The Mann–Whitney U test and Wilcoxon’s rank-sum test were used to analyze continuous variables, whereas the chi-square test or Fisher’s exact test were used to analyze categorical variables. Correlations between variables were analyzed using Spearman’s rank correlation. All statistical analyses were performed using STATA/SE software, version 16.1 (StataCorp; College Station, TX, USA). A two-tailed *p*-value < 0.05 indicated statistical significance.

## 3. Results

During the five-year study period, we enrolled a total of 53,088 severely injured patients with an ISS of ≥16 at 197 hospitals, and 2889 (5.4%) of them were pediatric patients aged <18 years old (Figure 1). The distribution of hospitals, stratified by the number of severely injured patients during the five-year study period, is shown in Figure 2. These patients were categorized into the following age groups: neonates/infants, 0; preschool children, 1–5; school-aged children, 6–12; adolescents, 13–17; adults, 18–64; and older adults, ≥65 years. The number of total and pediatric patients per hospital over the five years was 235 cases (1–1523) and 10 cases (1–114), respectively. The overall in-hospital mortality of SIPP per hospital was 4.5% (*n* = 129). 

### 3.1. Correlation Analysis

The results of the Spearman correlation analysis showed a significant correlation between the number of total patients and SIPP, per hospital, during the five-year period (R^2^ = 0.911; *p* < 0.001, Figure 3). 

The number of SIPP per hospital who underwent interhospital transportation, blood transfusion, TAE, craniotomy and/or craterization, thoracotomy, and celiotomy was correlated with the total number of severely injured patients per hospital (Table 1). Furthermore, the actual in-hospital mortality per hospital of SIPP was significantly correlated with the total number of patients per hospital (R^2^ = 0.095; *p* < 0.001, Figure 4).

### 3.2. Comparison of High-Volume Hospitals with the Low-Volume Hospitals

Among 197 eligible hospitals, 21 hospitals were excluded as no pediatric patients were admitted to these hospitals. Therefore, a total of 52,834 severely injured patients and 2889 SIPP from 176 hospitals were analyzed (Table 2). Of the 176 hospitals, 13 (7%) hospitals treated 12,437 (24% of all patients) severely injured patients and 750 SIPP (26% of all pediatric patients). 

A comparison of characteristics and outcomes between the two groups is summarized in Table 2. The total number of pediatric patients with severe injuries, interhospital transfer, polytrauma, blood transfusion, TAE, and craniotomy/craterization was higher in the high-volume hospital group than in the low-volume hospital group (2139 vs. 750 cases; 507 vs. 152 cases; 1531 vs. 503 cases; 358 vs. 112 cases; 127 vs. 46 cases; 283 vs.81 cases). There were no significant differences in actual in-hospital mortality (*p* = 0.246, 2.13 (0–8.33) vs. 0 (0–100)) and SMR values (*p* = 0.244, 0.31 (0–0.79) vs. 0 (0–4.87)), per hospital between the two groups; however, all the 13 hospitals with high volumes had an SMR of <1.0.

## 4. Discussion

To the best of our knowledge, this is the first nationwide study in Japan that evaluated the correlation between the volumes of total and pediatric hospital patient with severe injuries. Our study results showed that an increasing hospital volume of severely injured patients was significantly correlated with increased pediatric patient volume and decreased in-hospital mortality among SIPP. Moreover, high-volume hospitals with annual admissions of >150 severely injured patients accounted for only 7% of total hospitals and these hospitals treated three quarters of the total and SIPP patients who were registered in the JTDB. Finally, all high-volume hospitals had an SMR of <1.0, suggesting a favorable in-hospital mortality of SIPP.

Several studies have reported a volume–outcome relationship, where a higher hospital admission volume resulted in a lower mortality among injured, ICU admitted, and postoperative patients [7,8,9,10,13,14,15]. This study also showed that increasing hospital volumes resulted in a lower in-hospital mortality among SIPP. There might be several reasons explaining the fact that higher volume hospitals performed better. One reason might be that medical staff working at the high-volume hospitals gain a higher level of skills in trauma care because of the experience in managing a higher volume of severely injured patients, and a trauma team with higher proficiency is thus developed [7,16]. The next reason might be that the development of designated trauma teams for the management of severely injured patients in high-volume centers decreased mortality [16,17]. Particularly, it has been suggested that a trauma team centered around in-house trauma surgeons with much experiences and skills has a positive impact on the trauma care and outcomes among severely injured patients, although there is no definitive evidence [2,18,19]. The final reason might be that the change in infrastructure including medical resources at a high-volume center, such as new technological improvements and the introduction of development products, could lead to a decrease in mortality [7]. In this study, the increasing volume of severely injured patients, per hospital, was correlated with an increased number of urgent treatment, such as blood transfusion, TAE, and/or surgeons required. This result suggests that the increasing volume of patients who were admitted and who underwent urgent treatment, according to the centralization of severely injured patients, regardless of age, resulted in a lower in-hospital mortality. 

Although several studies showed that centralization of severely injured patients is effective in improving outcomes, the cut-off values of hospital volumes with which patients can gain a survival benefit maximally, remains under debate [7,8,9,10]. In this study, the high-volume hospitals with >150 severely injured patient admissions annually, included only 7% patients of all hospitals, although this number was lower than the required criteria of a level I trauma center in the United States [10]. Based on the present study, results indicated that an increasing hospital volume was correlated with a reducing in-hospital mortality and all high-volume hospitals had a survival benefit among SIPP as SMR <1.0. Therefore, centralizing severely injured patients to high-volume hospitals might have resulted in a decrease in-hospital mortality of SIPP in Japan. Moreover, in a country like Japan, where trauma centers specializing in pediatric care were not provided, centralization of severely injured patients regardless of age would be important for increasing hospital patient volume and decreasing in-hospital mortality.

This study showed that there were 2889 SIPP registered in the JTDB during the five-year study period and increasing hospital patient volumes were correlated with an increase in the number and proportion of SIPP with interhospital transportation. In summary, in Japan, a country with a low rate of SIPP and with geographical features in terms of the length of terrain and presence of many mountains, it was essential that trauma bypasses, including not only primary transport but also secondary transport, be constructed for promoting centralization of SIPP. Definitive trauma care that is provided within 1 h of injury has been associated with a significantly lower mortality risk [19]. Moreover, several reports showed that interhospital transportation among critically pediatric patients had a higher risk of a hypotension and hypoxia as well as delays in the decision-making process and definitive care than that among adults [20,21,22]. Therefore, timely and safe transportation of critically pediatric patients should be established with interhospital transport and hospital triage criteria, a transport team specialized in pediatric trauma care, and optimal transport methods such as ambulance or helicopter with a trained physician and a nurse [21,22,23]. 

This study had several limitations. The retrospective study design and some missing data in the JTDB impaired the precision of the analyses. Therefore, there was possibility for selection bias because all Japanese hospitals that treat trauma did not participate and register in the JTDB. Moreover, the number of participating hospitals differed across the study period. The severity of injured patients might differ in accordance with a participating hospital. Because a nationwide database with a large sample size was evaluated in this study, participants admitted in each year would represent a similar environment. In addition, we did not conduct analyses based on factors such as pediatric aged subgroups or the location of the accidental scene, type of injury, proportion of preventable trauma death, transport system, or cost of management. Several reports have shown that optimal treatment according to the abovementioned factors affects the outcomes. Moreover, we could not assess the differences in treatments among the facilities and the cause of death. Previous reports showed that mortality is predominantly associated with head and torso injuries with active bleeding in severely injured patients [24,25]. In particular, head injuries remain at a higher incident rate and mortality rate in Japanese SIPP [26]. Therefore, future nationwide studies with subclass analyses should be conducted to improve the outcome for SIPP. 

Furthermore, this study determined the correlation between total patient numbers and mortality of SIPP only. Moreover, this study showed that there were differences in actual mortality and SMR between high- and low-volume hospitals. In addition, there was a significant negative correlation between hospital volume and hospital mortality and all high-volume hospitals had SMR of <1.0. Therefore, additional studies, that demonstrate the association between hospital patient volume and the hospital mortality outcomes, and the optimal cut-off values of hospital patient volume for high-performance centers with favorable outcomes, are required. In addition, the disadvantages of interhospital transfer for severely injured patients such as adverse events during transportation as well as the delays in the decision-making process and provision of definitive care have been reported, therefore the criteria for severely injured patients that should be transferred to high-performance hospitals rather than stay at a local hospital, should be evaluated [27].

Finally, SIPP were defined as having ISS ≥16 in this study, because the patients with an ISS ≥16 is commonly used as the definition of severely injured patients. However, previous study reported that the predicted mortality of pediatric patients with an ISS of >25 was similar to adult patients with an ISS of >15 in the USA. Therefore, the optimal cut-off values for defining SIPP by using ISS should also be considered [28].

## 5. Conclusions

High-volume hospitals that admit >150 severely injured patients annually were associated with SIPP survival benefit. Centralizing severely injured patients, regardless of age, to higher volume hospitals may contribute the survival benefit of SIPP. 

## Figures and Tables

**Figure 1 jcm-10-01422-f001:**
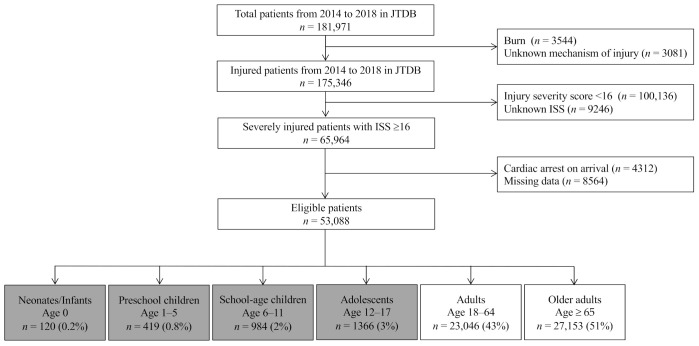
Flow diagram of study patient disposition. JTDB, Japanese Trauma Data Bank; ISS, Injury Severity Score.

**Figure 2 jcm-10-01422-f002:**
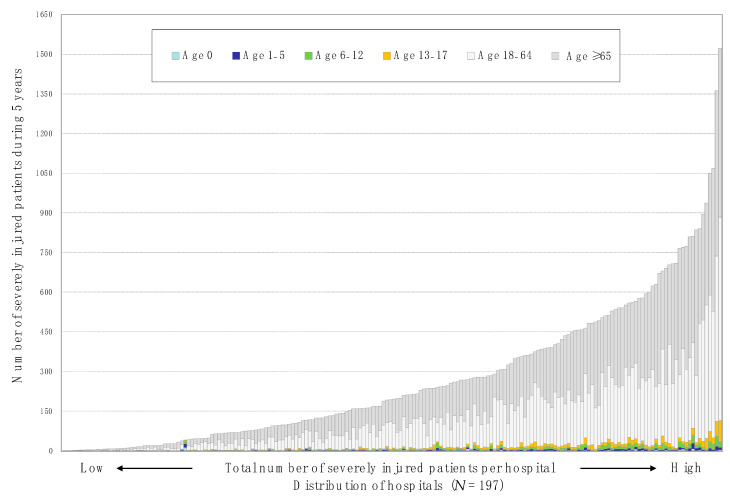
Distribution of hospitals stratified by the number of severely injured patients per hospital during the 5-year study period.

**Figure 3 jcm-10-01422-f003:**
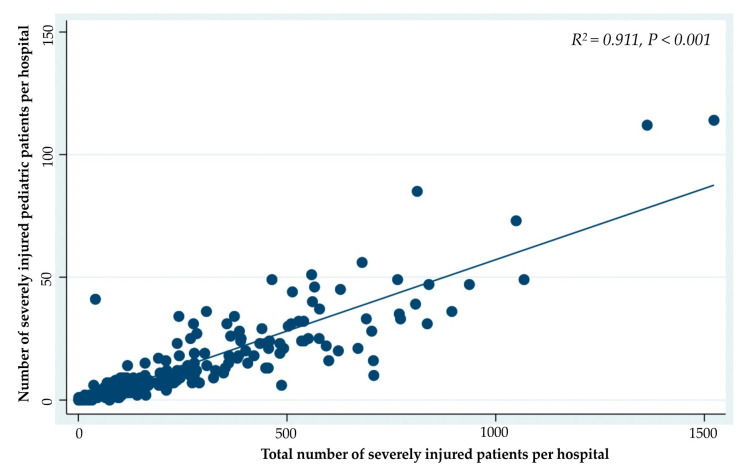
Correlation between the number of total patients and the number of pediatric patients with severe injuries, per hospital, during the 5-year study period.

**Figure 4 jcm-10-01422-f004:**
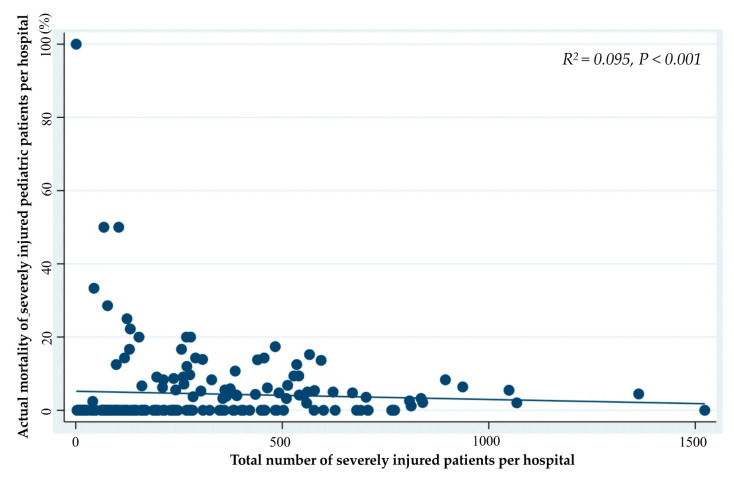
Correlation between the total numbers of severely injured patients and actual mortality of severely injured pediatric patients, per hospital, during the 5-year study period.

**Table 1 jcm-10-01422-t001:** Correlation between the number of severely injured patients and the number of severely injured pediatric patients who underwent interhospital transportation and urgent treatment.

	R^2^	*p*-Value
**Correlation between the Number of Total Patients and the Number of Severely Injured Pediatric Patients Who Underwent**		
Interhospital transportation	0.453	<0.001
Urgent treatment		
Blood transfusion within 24 h	0.499	<0.001
Transcatheter arterial embolization	0.286	<0.001
Craniotomy and/or craterization	0.435	<0.001
Thoracotomy	0.117	<0.001
Celiotomy	0.325	<0.001

**Table 2 jcm-10-01422-t002:** Comparison of variables between high- and low-volumes hospitals.

Variables	All Hospitals	High-Volume Hospitals	Low-Volume Hospitals	*p*-Value
	(176 Hospitals)	(13 Hospitals)	(163 Hospitals)	
Total number of severely injured patients	52,834	40,397	12,437	
Total number per institution, (median, min–max)	235 (1–1523)	840 (765–1523)	210 (1–708)	<0.001
0 year	0 (0–12)	1 (0–7)	0 (0–12)	0.005
1–5 years	1(0–22)	9 (1–22)	1 (0–14)	<0.001
6–12 years	3 (0–39)	17 (4–39)	3 (0–24)	<0.001
13–17 years	5 (2–80)	24 (10–80)	4 (0–29)	<0.001
18–64 years	93 (0–770)	435 (238–770)	89 (0–374)	<0.001
≥65 years	113 (0–639)	461 (358–639)	99 (0–43)	<0.001
**PEDIATRIC**				
Total number of severely injured pediatric patients	2889	2139	750	
Total number of severely injured pediatric patients per institution, (median, min–max)	10 (1–114)	47 (31–114)	9 (1–56)	<0.001
Blunt trauma, number of pediatric patients per institution, (median, min–max)	10 (1–113)	47 (31–113)	9 (1–56)	<0.001
Transport type, number of pediatric patients per institution, (median, min–max)				
Ambulance	8 (0–96)	28 (8–96)	7 (0–43)	<0.001
Ambulance with physician	0 (0–31)	3 (0–31)	0 (0–24)	<0.001
Helicopter with physician	0 (0–61)	9 (0–61)	0 (0–12)	<0.001
Transfer process, number of pediatric patients per institution, (median, min–max)				
From the scene	8 (3–109)	38 (20–109)	7 (0–49)	<0.001
From the other hospital	2 (0–36)	11 (3–36)	1 (0–28)	<0.001
Injury region, number of pediatric patients per institution, (median, min–max)				
Polytrauma	3 (0–64)	17 (4–64)	3 (0–24)	<0.001
Head injury with AIS ≥3	7 (0–73)	27 (19–73)	7 (0–44)	<0.001
Facial injury with AIS ≥3	0 (0–6)	0 (0–6)	0 (0–4)	0.022
Neck injury with AIS ≥3	0 (0–2)	0 (0–2)	0 (0–1)	0.231
Chest injury with AIS ≥3	3 (0–63)	19 (5–63)	3 (0–25)	<0.001
Abdominal and pelvic injury with AIS ≥3	1 (0–18)	7 (1–18)	1 (0–12)	<0.001
Spinal injury with AIS ≥3	0 (0–9)	4 (0–9)	0 (0–9)	<0.001
Upper extremity injury with AIS ≥3	0 (0–10)	0 (0–10)	0 (0–3)	<0.001
Lower extremity injury with AIS ≥3	1 (0–35)	7 (1–35)	1 (0–12)	<0.001
ISS score range, total number of severely injured pediatric patients				
16–25	2018	1491	527	<0.001
>25	871	648	223	<0.001
Revised trauma score per institution, (median, min–max)	7.15 (4.00–7.84)	7.12 (6.39–7.06)	7.16 (4.00–7.84)	0.874
TRISS score per institution, (median, min–max)	0.93 (0.48–0.99)	0.93 (0.79–0.95)	0.93 (0.48–0.99)	0.516
Urgent treatment, number of pediatric patients per institution, (median, min–max)				
Blood transfusion within 24 h	1 (0–21)	7 (1–21)	1 (0–17)	<0.001
Urgent transcatheter arterial embolization	0 (0–12)	2 (0–12)	0 (0–10)	0.004
Craniotomy and/or craterization	1 (0–15)	6 (0–15)	1 (0–11)	<0.001
Thoracotomy	0 (0–2)	0 (0–2)	0 (0–1)	0.068
Celiotomy	0 (0–5)	2 (0–5)	0 (0–5)	<0.001
Predicted mortality per institution, (median, min–max)	6.74 (1.04–51.57)	7.18 (4.67–20.89)	6.62 (1.04–51.57)	0.516
Actual in-hospital mortality per institution, (median, min–max)	0 (0–100)	2.13 (0–8.33)	0 (0–100)	0.246
SMR score range per institution, (median, min–max)	0 (0–4.87)	0.31 (0–0.79)	0 (0–4.87)	0.244
Number of institutions stratified by SMR score range per institution, (frequency, %)				
0.00–1.00	158 (90)	13 (100)	145 (89)	0.206
>1.00	18 (10)	0 (0)	18 (11)	

AIS: Abbreviated Injury Scale, ISS: Injury Severity Score, TRISS: Trauma and Injury Severity Score, Min: minimum, Max: maximum, SMR: standardized mortality ratio.

## Data Availability

The approving authority for data access was the Japanese Association for the Surgery of Trauma (Trauma Registry Committee).

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
