# Peer review of "Correlation between Hospital Volume of Severely Injured Patients and In-Hospital Mortality of Severely Injured Pediatric Patients in Japan: A Nationwide 5-Year Retrospective Study"

_jcm, 2021, doi:10.3390/jcm10071422_

Round 1

Reviewer 1 Report

This work highlights the current practice of management of severely injured pediatric patients (SIPP) in Japan. The authors state that in lack of dedicated pediatric trauma centers in their country, the best outcome of pediatric patients is achieved when patients are treated in high volume centers and they back up their statement by giving the information that in high volume centers the standardized mortality ratio (SMR) of SIPP was below 1.0 score range.

This result was to be expected but there is a number of problems with authors statement.

First, although many significant correlations can be found in the manuscript (even the frequency of very low number of thoracotomies …) , their statement of decreased mortality ratio in high volume centers is not backed up by significance. So mortality or SMR values were not significantly lower in high volume centers.

Furthermore, the authors are not giving the absolute numbers of fatalities, the reader needs to calculate himself, which is not interesting and time consuming. Mortality is stated by 4,5% in 2889 SIPP, which is around 130 fatalities in 5 years, or 26 per year.

Recent paper by Brown JB et al showed that ISS of above 25 is better predictive of outcome of SIPP than ISS of 16.

I miss the correlation of the ISS with mortality.

Also, the absolute numbers of SIPP with ISS >45 should be given in table 2, not only the median and range, as the reader cannot be sure why the results are as they are!

Line 122 the absolute number: 2889 pediatric patients  should be added and the value 5,4% given in parenthesis.

Line 71 and 72 and 111: it is probably Japanese Association … not Japanese Correlation

Line 139: it must be better explained what the “number and frequency” mean in this sentence as it is given further in the results and it is also not very clear especially what “frequency” means/refers to!

Lines 134 – 138 : these 2 sentences have the same meaning and though they are completely opposite concerning the result. Please clarify, because they cannot stand like this in the manuscript.

Table 2 is giving detailed analysis, but receives almost no attention of the authors in the Results or Discussion. Mortality is mainly associated to Head or Chest injury in SIPP. The authors do not comment on this in their survey.

There is no real evidence that low-volume centers performed less good than high-volume centers.

Page 7 Table 2 SMR score range per institution – per institution is written 2 times – delete one!

Line 246 : add “with” before the word increased at the beginning of the line

Use throughout the manuscript: severely injured pediatric patients, not “pediatric severely injured patients”

Joshua B Brown, Mark L Gestring, Christine M Leeper, Jason L Sperry, Andrew B Peitzman, Timothy R Billiar, Barbara A Gaines. The value of the injury severity score in pediatric trauma: Time for a new definition of severe injury? J Trauma Acute Care Surg. 2017 Jun;82(6):995-1001.

doi:10.1097/TA.0000000000001440.

Author Response

Reviewer’s comments (in blue) and Answers (in black) follow.

We wish to express our deep appreciation for the valuable comments from the Reviewer regarding our manuscript. We believe that our manuscript has greatly benefited from these comments.

Reviewer: 1

  1. This work highlights the current practice of management of severely injured pediatric patients (SIPP) in Japan. The authors state that in lack of dedicated pediatric trauma centers in their country, the best outcome of pediatric patients is achieved when patients are treated in high volume centers and they back up their statement by giving the information that in high volume centers the standardized mortality ratio (SMR) of SIPP was below 1.0 score range. This result was to be expected but there is a number of problems with authors statement. First, although many significant correlations can be found in the manuscript (even the frequency of very low number of thoracotomies …), their statement of decreased mortality ratio in high volume centers is not backed up by significance. So mortality or SMR values were not significantly lower in high volume centers.

Response: We thank the Reviewer for this valuable comment. We agree that this point, as you have suggested, is the dominant limitation of this study. Therefore, we have added the comparison of actual mortality and SMR to the Results section, and have added this as a limitation in the Discussion sections as follows:

Results section

”A comparison of characteristics and outcomes between the two groups is summarized in Table 2. The total number of pediatric patients with severe injuries, interhospital transfer, polytrauma, blood transfusion, TAE, and craniotomy/craterization was higher in the high-volume hospital group than in the low-volume hospital group (2139 vs. 750 cases; 507 vs. 152 cases; 1531 vs. 503 cases; 358 vs. 112 cases; 127 vs. 46 cases; 283 vs. 81 cases). There were no significant differences in actual in-hospital morality [p=0.246, 2.13 (0–8.33) vs. 0 (0–100)] and SMR values [p=0.244, 0.31 (0–0.79) vs. 0 (0–4.87)], per hospital, between the two groups; however, the 13 hospitals with high volumes had an SMR of <1.0.”

Discussion section

”Furthermore, this study determined the correlation between total patient numbers and mortality of SIPP only. Moreover, this study showed that there were differences in actual mortality and SMR between high- and low-volume hospitals. In addition, there was a significant negative correlation between hospital volume and hospital mortality and all high-volume hospitals had an SMR of <1.0. Therefore, additional studies, that demonstrate the association between hospital patient volume and the hospital mortality outcomes, and the optimal cut-off values of hospital patient volumes for high-performance centers with favorable outcomes, are required.”

  1. Furthermore, the authors are not giving the absolute numbers of fatalities, the reader needs to calculate himself, which is not interesting and time consuming. Mortality is stated by 4,5% in 2889 SIPP, which is around 130 fatalities in 5 years, or 26 per year.

Response: We thank the Reviewer for this comment. As requested, we have added the absolute numbers of fatalities as follows:

“The overall in-hospital mortality of SIPP, per hospital, was 4.5% (n=129).”

  1. Recent paper by Brown JB et al showed that ISS of above 25 is better predictive of outcome of SIPP than ISS of 16. I miss the correlation of the ISS with mortality. Also, the absolute numbers of SIPP with ISS >45 should be given in table 2, not only the median and range, as the reader cannot be sure why the results are as they are!

Response: We thank the Reviewer for this valuable comment. We agree that additional information regarding the definition of severely injured pediatric patients by ISS is required. Therefore, we have added the total number of patients with ISS 16-25/ >25 to Table 2 and discussed this in the Discussion section as a limitation of our study in accordance with the reference you recommended as follows:

Table 2

ISS score range, total number of severely injured pediatric patients

16–25

2018

1491

527

<0.001

>25

871

648

223

<0.001

Discussion section

” Finally, SIPP were defined as having an ISS of ³16 in this study, because an ISS of ³16 is commonly used as the definition of severely injured patients. However, a previous study reported that the predicted mortality of pediatric patients with an ISS of >25 was similar to adult patients with as ISS of >15 in the USA. Therefore, the optimal cut-off values for defining SIPP by using ISS should also be considered [28].”

References

  1. Brown, B.B.; Gestring, M.L.; Leeper, C.M.; Sperry, J.L.; Peitzman, A.B.; Billiar, T.R.; Gaines, B.A. The value of the injury severity score in pediatric trauma: Time for a new definition of severe injury? J Trauma Acute Care Surg 2017, 82, 995–1001.

  1. Line 122 the absolute number: 2889 pediatric patients should be added and the value 5,4% given in parenthesis.

Response: We thank the reviewer for this suggestion. We have revised the Abstract and Result section as follows:

“... 2889 (5.4%) were pediatric patients aged <18 years old.”

  1. Line 71 and 72 and 111: it is probably Japanese Association … not Japanese Correlation

Response: We thank the Reviewer for this comment. The error has been corrected.

  1. Line 139: it must be better explained what the “number and frequency” mean in this sentence as it is given further in the results and it is also not very clear especially what “frequency” means/refers to!

Response: We thank the Reviewer for this comment. We used the term “frequency” to indicate the “incident rate of SIPP” or “occurrence rate of urgent intervention”. However, we agree with this suggestion, we have thus revised the expression regarding frequency throughout this paper.

  1. Lines 134 – 138 : these 2 sentences have the same meaning and though they are completely opposite concerning the result. Please clarify, because they cannot stand like this in the manuscript.

Response: We thank the Reviewer for this comment. We think that a misunderstanding regarding “frequency” arose because we used this term incorrectly as mentioned above in comment 6. Therefore, we have revised these two sentences as follows:

“In contrast, the rate of SIPP incidence, per hospital, was not significantly correlated with the total number of patients with severe injury per hospital (R2 = 0.004; P=0.406).”

  1. Table 2 is giving detailed analysis, but receives almost no attention of the authors in the Results or Discussion.

Response: We thank the Reviewer for this comment. As we mentioned above your comment 1, we have added detailed results to the Results section and information to the Discussion section as follows:

Results section

”A comparison of characteristics and outcomes between the two groups is summarized in Table 2. The total number of pediatric patients with severe injuries, interhospital transfer, polytrauma, blood transfusion, TAE, and craniotomy/craterization was higher in the high-volume hospital group than in the low-volume hospital group (2139 vs. 750 cases; 507 vs. 152 cases; 1531 vs. 503 cases; 358 vs. 112 cases; 127 vs. 46 cases; 283 vs. 81 cases). There were no significant differences in actual in-hospital morality [p=0.246, 2.13 (0–8.33) vs. 0 (0–100)] and SMR values [p=0.244, 0.31 (0–0.79) vs. 0 (0–4.87)], per hospital, between the two groups; however, the 13 hospitals with high volumes had an SMR of <1.0.”

Discussion section

“Furthermore, this study determined the correlation between total patient numbers and mortality of SIPP only. Moreover, this study showed that there were differences in actual mortality and SMR between high- and low-volume hospitals. In addition, there was a significant negative correlation between hospital volume and hospital mortality and all high-volume hospitals had an SMR of <1.0. Therefore, additional studies, that demonstrate the association between hospital patient volume and the hospital mortality outcomes, and the optimal cut-off values of hospital patient volumes for high-performance centers with favorable outcomes, are required.”

  1. Mortality is mainly associated to Head or Chest injury in SIPP. The authors do not comment on this in their survey.

Response: We thank the Reviewer for this insightful suggestion. We agree that additional discussion points regarding your suggestion, are required. Therefore, we have added a limitation of this study as follows:

Discussion section

“Previous reports showed that mortality is predominantly associated with head and torso injuries with active bleeding in severely injured patients [24,25]. In particular, head injuries remain at a higher incident and mortality rate in Japanese SIPP [26]. Therefore, future nationwide studies with subclass analyses should be conducted to improve the outcomes for SIPP.”

References

  1. Brohi, K.; Gruen, R.L.; Holcomb, J.B. Why are bleeding trauma patients still dying? Intensiv Care Med 2019, 45, 709–711.
  2. Brazinova, A.; Rehorcikova, V.; Taylor, M.S.; Buckova, V.; Majdan, M.; Psota, M.; Peeters, W.; Feigin, V.; Theadom, A.; Holkovic, L.; Synnot, A.. Epidemiology of traumatic brain injury in Europe: A living systematic review. J Neurotrauma 2018, 33, 1–30.
  3. Toida, C.; Muguruma, T.; Gakumazawa, M.; Shinohara, M.; Abe, T.; Takeuchi, I.; Morimura, N. Age- and severity-related in-hospital mortality trends and risks of severe traumatic brain injury in Japan: A nationwide 10-year retrospective study. J Clin Med 2021, 10, 1072.

  1. There is no real evidence that low-volume centers performed less good than high-volume centers.

Response: We thank the Reviewer for this comment. We agree with the suggestion. This study determined the correlation between the total number of patients and mortality of SIPP only. Therefore, we consider an additional study that is required to show the association between hospital patient volumes and the hospital mortality outcomes, and the optimal cut-off values of hospital patients volumes for high-performance centers with favorable outcomes. Accordingly, we have added this content to the manuscript as a limitation as follows:

” Furthermore, this study determined the correlation between total patient numbers and mortality of SIPP only. Moreover, this study showed that there were differences in actual mortality and SMR between high- and low-volume hospitals. In addition, there was a significant negative correlation between hospital volume and hospital mortality and all high-volume hospitals had an SMR of <1.0. Therefore, additional studies, that demonstrate the association between hospital patient volume and the hospital mortality outcomes, and the optimal cut-off values of hospital patient volumes for high-performance centers with favorable outcomes, are required.”

  1. Page 7 Table 2 SMR score range per institution – per institution is written 2 times – delete one!

Response: We thank the Reviewer for this comment. The error has been corrected.

  1. Line 246 : add “with” before the word increased at the beginning of the line

Response: We thank the Reviewer for this comment. The error has been corrected.

  1. Use throughout the manuscript: severely injured pediatric patients, not “pediatric severely injured patients”

Response: We thank the Reviewer for this comment. We have changed the term from “pediatric severely injured patients” to “severely injured pediatric patients (SIPP)”, throughout the manuscript.

Reviewer 2 Report

Well written paper and of interest to field. 

Abstract (and throughout paper): Recommend including quantifiable values and specific comparisons where relevant, e.g. for interhospital transfers, % with urgent treatment, mortality rates by hospital size, etc., and at least some numerical values for severe injury, polytrauma, transfusion, craniotomy/craniectomy between centers, to guide the reader. In the conclusions the authors state "high volume hospitals had a survival benefit... " - this should be quantified.

Is there a way to establish at what point/severity cutoff that patients should be transferred to larger hospitals rather than stay at a local hospital, or should severe patients always be transferred to larger hospitals? This incurs a cost to the healthcare system/resources and to the patient, so it should be discussed.

Author Response

Reviewer’s comments (in blue) and Answers (in black)

We wish to express our deep appreciation for the valuable comments from the Reviewer regarding our manuscript. We believe that our manuscript has greatly benefited from these comments.

Reviewer: 2

  1. Abstract (and throughout paper): Recommend including quantifiable values and specific comparisons where relevant, e.g. for interhospital transfers, % with urgent treatment, mortality rates by hospital size, etc., and at least some numerical values for severe injury, polytrauma, transfusion, craniotomy/craniectomy between centers, to guide the reader.

Response: We thank the Reviewer for this valuable comment. As requested, we have added information regarding the comparisons of the two hospital groups to the Results section. As the number of words was limited in the Abstract, we could not show all these data in the Abstract section.

Abstract

” The total number of SIPP, requiring urgent treatment, was higher in the high-volume than in the low-volume hospital group. No significant differences in actual in-hospital morality [p=0.246, 2.13 (0–8.33) vs. 0 (0–100)] and standardized mortality ratio (SMR) values [p=0.244, 0.31 (0–0.79) vs. 0 (0–4.87)] were observed between the two groups; however, the 13 high-volume hospitals had an SMR of <1.0.”

Results section

” A comparison of characteristics and outcomes between the two groups is summarized in Table 2. The total number of pediatric patients with severe injuries, interhospital transfer, polytrauma, blood transfusion, TAE, and craniotomy/craterization was higher in the high-volume hospital group than in the low-volume hospital group (2139 vs. 750 cases; 507 vs. 152 cases; 1531 vs. 503 cases; 358 vs. 112 cases; 127 vs. 46 cases; 283 vs. 81 cases). There were no significant differences in actual in-hospital morality [p=0.246, 2.13 (0–8.33) vs. 0 (0–100)] and SMR values [p=0.244, 0.31 (0–0.79) vs. 0 (0–4.87)], per hospital, between the two groups; however, the 13 hospitals with high volumes had an SMR of <1.0.”

  1. In the conclusions the authors state "high volume hospitals had a survival benefit... " - this should be quantified. Is there a way to establish at what point/severity cutoff that patients should be transferred to larger hospitals rather than stay at a local hospital, or should severe patients always be transferred to larger hospitals? This incurs a cost to the healthcare system/resources and to the patient, so it should be discussed.

Response: We thank the Reviewer for this valuable comment. As requested, we have added the optimal cut-off values and specific patients who experienced a survival benefit because of transfer to a high performance center to the Discussion section as follows:

Discussion section

“Therefore, additional studies, that demonstrate the association between hospital patient volume and the hospital mortality outcomes, and the optimal cut-off values of hospital patient volumes for high-performance centers with favorable outcomes, are required. In addition, the disadvantages of interhospital transfer for severely injured patients such as adverse events during transportation as well as the delays in the decision-making process and provision of definitive care have been reported, therefore the criteria for severely injured patients that should be transferred to high performance hospitals rather than stay at a local hospital, should be evaluated [27].”

References

  1. Adzemovic, T.; Murray, T.; Jenkins, P.; Ottosen, J.; Iyegha, U.; Raghavendran, K.; Napolitano, L.M.; Hemmila, M.R.; Gipson, J.; Park, P.; Tignanelli, C.J. Should they stay or should they go? Who benefits from interfacility transfer to a higher-level trauma center following initial presentation at a lower-level trauma center. J Trauma Acute Care Surg 2019, 86, 952–960.

Round 2

Reviewer 1 Report

This is a much improved submission. However, there are a few more minor changes to be made:

Line 23: instead of fullstop (.) I think it should be comma (,) and "The" written small (the)

Line 23: "occurence rate" - these words are now occurring throughout the manuscript. It is not clear to me: please if the authors compared numbers with numbers, then they should delete "occurence rate" from the manuscript. If the authors compared the occurence rate with occurence rate than maybe it could be appropriate but it is definitely confusing. And I don't really see the way how comparing of numbers with occurrence rate should be appropriate. 

Lines 135-138: these 2 first sentences at the beginning of Correlation analysis are still mixed up. In the first sentence the authors write that there is a significant correlation between the total number of patients per hospital and the total number of SIPP per hospital and they give the correlation R2=0.911 - Figure 3). In the 2nd sentence the authors write that the rate of SIPP incidence per hospital (dont understand what this is??? - should stay the number of SIPP per hospital) was not significantly correlated with the total number of severely injured patients per hospital and they give the correlation R2=0.004).

The Figure 3 showes however on the X-axis "the total number of severely injured patients" which in the second sentance is given as not significant ... this is mixed up and I would appreciate if the authors would make this clear because this is very confusing.  They have clear data, but they are making their own data complicated - I dont understand why!

Line 167: mor(t)ality - "t" is missing

Lines 259-260: The sentence as it stays is not supported by the data: the last part of the sentence should be deleted - DELETE "and decreases in-hospital mortality among SIPP". The authors state anyway in the next sentence that there was a survival benefit for SIPP in high volume hospitals which is supported by the data. 

Author Response

Reviewer’s comments (in blue) and Answers (in black) follow.

We wish to express our deep appreciation for the valuable comments from the Reviewer regarding our manuscript. We believe that our manuscript has greatly benefited from these comments.

Reviewer: 1

  1. Line 23: instead of fullstop (.) I think it should be comma (,) and "The" written small (the)

Response: We thank the reviewer for providing this valuable comment. Accordingly, we have revised this point.

  1. Line 23: "occurence rate" - these words are now occurring throughout the manuscript. It is not clear to me: please if the authors compared numbers with numbers, then they should delete "occurence rate" from the manuscript. If the authors compared the occurence rate with occurence rate than maybe it could be appropriate but it is definitely confusing. And I don't really see the way how comparing of numbers with occurrence rate should be appropriate.

Response: We thank the reviewer for this comment. We agree that the term “occurrence rate” was not clear to the readers and should be deleted. We have discussed about pediatric mortality by focusing on the number of severely injured patients per hospital; therefore, we have deleted the description relating to the occurrence rate.  

  1. Lines 135-138: these 2 first sentences at the beginning of Correlation analysis are still mixed up. In the first sentence the authors write that there is a significant correlation between the total number of patients per hospital and the total number of SIPP per hospital and they give the correlation R2=0.911 - Figure 3). In the 2nd sentence the authors write that the rate of SIPP incidence per hospital (dont understand what this is??? - should stay the number of SIPP per hospital) was not significantly correlated with the total number of severely injured patients per hospital and they give the correlation R2=0.004).

Response: We thank the reviewer for this comment. We agree that the term “occurrence rate” was not clear to the readers and should be deleted. We have discussed pediatric mortality by focusing on the number of severely injured patients per hospital; therefore, we have deleted the second sentence you indicated.

  1. The Figure 3 showes however on the X-axis "the total number of severely injured patients" which in the second sentance is given as not significant ... this is mixed up and I would appreciate if the authors would make this clear because this is very confusing. They have clear data, but they are making their own data complicated - I don’t understand why!

Response: We thank the reviewer for this comment. Because we had incorrectly pointed out the location where Figure 3 should be inserted, it was difficult to understand the results that the Figure 3 shows correctly. Therefore, we have inserted the Figure 3 into the results section close to its citation.

  1. Line 167: mor(t)ality - "t" is missing

Response: We thank the reviewer for this comment. The error has been corrected.

  1. Lines 259-260: The sentence as it stays is not supported by the data: the last part of the sentence should be deleted - DELETE "and decreases in-hospital mortality among SIPP". The authors state anyway in the next sentence that there was a survival benefit for SIPP in high volume hospitals which is supported by the data.

Response: We thank the reviewer for this valuable comment. We agree with this point, as you have suggested. Therefore, we have revised the conclusions, which are now supported by our data as follows:

High-volume hospitals that admit >150 severely injured patients annually were associated with SIPP survival benefit. Centralizing severely injured patients, regardless of age, to higher volume hospitals may contribute to the survival benefit of SIPP.
